# Development and characterization of an antibody that recognizes influenza virus N1 neuraminidases

Nan Chen[1], Renxi Wang[2], Wanlu Zhu[1,3], Xiangjun Hao[1,3], Jing Wang[1], Guojiang Chen[1], ChunXia Qiao[1], Xinying Li[1], Chenghua Liu[1], Beifen Shen[1], Jiannan Feng[1], Lihui Chai[3]*, Zuyin Yu[4]*, He Xiao[1]*

1 State Key Laboratory of Toxicology and Medical Countermeasures, Beijing Institute of Pharmacology and Toxicology, Beijing, China, 2 Laboratory of Brain Disorders, Collaborative Innovation Center for Brain Disorders, Beijing Institute of Brain Disorders, Capital Medical University, Ministry of Science and Technology, Beijing, China, 3 Joint National Laboratory for Antibody Drug Engineering, The First Affiliated Hospital, School of Medicine, Henan University, Kaifeng, China, 4 Department of Experimental Hematology and Biochemistry, Beijing Key Laboratory for Radiobiology, Beijing Institute of Radiation Medicine, Beijing, China

* xiaoheysy@163.com (HX); yuzy79@163.com (ZY); clh0301@henu.edu.cn (LC)

**Data Availability Statement:** The raw data of flow cytometry experiments is available from the figshare database: https://figshare.com/s/22b41e73e44bcc7d11e2. The DNA sequences of

## Abstract

Influenza A viruses (IAVs) continue to pose a huge threat to public health, and their prevention and treatment remain major international issues. Neuraminidase (NA) is the second most abundant surface glycoprotein on influenza viruses, and antibodies to NA have been shown to be effective against influenza infection. In this study, we generated a monoclonal antibody (mAb), named FNA1, directed toward N1 NAs. FNA1 reacted with H1N1 and H5N1 NA, but failed to react with the NA proteins of H3N2 and H7N9. In vitro, FNA1 displayed potent antiviral activity that mediated both NA inhibition (NI) and blocking of pseudo-virus release. Moreover, residues 219, 254, 358, and 388 in the NA protein were critical for FNA1 binding to H1N1 NA. However, further validation is necessary to confirm whether FNA1 mAb is indeed a good inhibitor against NA for application against H1N1 and H5N1 viruses.

## Introduction

Influenza A viruses (IAVs) are highly contagious pathogens [1], which cause seasonal flu epidemics almost every year and represent significant public health burdens globally [2]. The IAV envelope surface contains two important membrane glycoproteins, including hemagglutinin (HA) and neuraminidase [3, 4], which are essential determinants of virus infectivity, transmissibility, pathogenicity, and major antigenicity [5].

Based on antigenic distinction, influenza A virus is divided into 18 HA (H1 to H18) and 11 NA subtypes (N1 to N11) [6]. The viral infection process is initiated by HA molecules [7], which mediate the binding of influenza viruses to sialic acid receptors on the surface of host cells [8, 9], and are the targets for neutralizing antibody responses [10]. However, due to its

the FNA1 heavy and light chain variable regions have been deposited at GenBank with the accession numbers of PP483115 (heavy chain) and PP483116 (light chain).

**Funding:** The author(s) received no specific funding for this work.

**Competing interests:** The authors have declared that no competing interests exist.

immunodominance [11, 12], HA undergoes antigenic drift, which can lead to a dramatic loss in the effectiveness of neutralizing antibodies. Although mutations also occur in NA protein, the rate of NA antigenic drift is comparatively slower [13, 14] and more conserved than that of HA between subtypes [15]. Therefore, NA is a potential target for more broadly protective vaccines or antibodies [16].

As a sialidase, NA cleaves terminal sialic acids on glycans expressed on the host cell surface, enabling the release and spread of newly synthesized viruses [17, 18]. Additionally, NA can break down mucins in the respiratory tract at the early stage of infection, allowing the virus to more efficiently penetrate the mucus layers, thereby playing an important role during infection [19, 20].

Currently, oseltamivir is the most common neuraminidase inhibitor (NAI) and has been approved for the treatment of IAVs worldwide [21, 22]. However, with the widespread use of oseltamivir, the emergence and global transmission of resistant variants have become a concern [23], necessitating the development of safer and more effective NAIs. Monoclonal antibodies are a viable treatment for influenza [24], with increasing studies indicating that NA-inhibiting antibodies can provide protection from influenza viruses and mitigate the severity of the disease [25, 26]. NI mAbs can inhibit NA enzymatic activity via direct binding or steric hinderance of the active site [14, 27], thereby preventing virion budding and spread from the host cell. Moreover, anti-NA antibodies that bind to NA at the surface of infected cells may aid in the clearance of the viruses through Fc–FcγR interactions to activate effector cells [28].

Here, we describe the characterization of an antibody for N1 NAs that was identified from a human antibody phage display library. The antibody, termed FNA1, binds to H1N1 and H5N1 NAs and inhibits the enzymatic activity of NA. We demonstrated that FNA1 mAb can restrict pseudovirus particle release using the H5N1 pseudovirus infection assay. Finally, site-directed mutagenesis was used to preliminarily identify the antigen epitope that FNA1 mAb recognized, and we confirmed several residues that contributed to the H1N1 NA interactions with FNA1.

## Materials and methods

### Cells and recombinant proteins

Madin–Darby canine kidney (MDCK) cells and human embryonic kidney 293T cells were obtained from ATCC (USA) and maintained in Dulbecco's Modified Eagle's Medium (complete DMEM, Gibco, USA) supplemented with 10% fetal bovine serum (FBS, Gibco). These cells were incubated at 37°C under 5% $CO_2$.

Recombinant NA proteins derived from A/California/04/2009 (H1N1, CA/09), A/Hong Kong/4801/2014 (H3N2), A/Anhui/1/2005 (H5N1, Anh05), and A/Anhui/1/2013 (H7N9) were obtained from Sino Biological (China, 11058-V08B, 40569-V07H, 11676-V08B, and 40108-V07H, respectively).

### Library panning and monoclonal phage ELISA

A human single-chain variable fragment (scFv) antibody phage library constructed in our laboratory was used to select specific binders against H1N1 NA (unpublished data). Based on computer-aided molecular design, we designed the scFv library using the retrieved antibody variable region sequences as templates. Panning and screening were performed as described previously [29]. Briefly, immunotubes (Bio-view shine, China) were coated with CA/09 NA at 4°C overnight. Antibody library phages were incubated with NA for 1 h and unbound phages were removed by washing with PBST (0.1% Tween 20 in phosphate buffered saline [PBS]) 20 times. The bound phages were eluted using 0.1 M HCl–glycine for 10 min and then used to

infect E. coli TG1 cells. TG1 cells were then plated on a 2 × YT/glucose/chloramphenicol (2YT-GC) agar plate and incubated overnight. The colonies were collected by scraping, and the scraped cells were inoculated in 2YT-GC medium and shaken at (220 rpm) until the optical density at 600 nm ($OD_{600}$) reached 0.6. The culture was super-infected with M13KO7 helper phages and grown in 2 × YT/chloramphenicol/kanamycin (2YT-CK) medium overnight at 28˚C. Phage particles in the culture supernatant were precipitated using polyethylene glycol 8000 (PEG 8000) on ice and resuspended in PBS. The amplified phages were used for the next round of panning.

After the third round of panning, 88 colonies were randomly selected, inoculated into 1 mL of 2YT-GC medium in 96-deep-well plates (Thermo Fisher Scientific, USA, 278743), and incubated at 37˚C until the $OD_{600}$ reached approximately 0.6. Subsequently, M13KO7 helper phages were added to the plates and incubated overnight at 28˚C. The plates were centrifuged, and the phage supernatant was used for enzyme-linked immunosorbent assay (ELISA). Briefly, a 96-well high-binding microplate (Corning, USA, 9018) was coated with 2 µg/mL CA/09 NA overnight at 4˚C. After blocking with 5% skimmed milk in PBST, the plate was incubated with 100 µL of phage supernatant at 37˚C for 1 h, before washing three times with PBST. Subsequently, horseradish peroxidase (HRP)-conjugated anti-M13 antibody (1:1000, Abcam, UK, ab50370) was added and incubated at 37˚C for 30 min. Colorimetric detection was performed using a tetramethylbenzidine (TMB) substrate solution (ComWin, China, CW0050S). The reaction was terminated by the addition of 1 M $H_2SO_4$, and the absorbance was measured at 450 nm using a microplate reader (Molecular Devices, USA). Clones showing specific binding activity were subjected to DNA sequencing. Finally, we selected an scFv with good activity and named it FNA1 for the presentation of the results.

## MAb production

To convert the selected scFv into IgG format, the variable region sequences of the heavy chain (VH) and light chain (VL) were synthesized by Sangon Biotech Company (China). The resulting VH and VL sequences were cloned into the pFRT-IgG1κ vector to construct the recombinant expression plasmid, pFRT-IgG1κ-FNA1. The pFRT-IgG1κ-FNA1 plasmid was introduced into ExpiCHO cells using the ExpiFectamine CHO transfection kit (Thermo Fisher Scientific, USA, A29133) following the manufacturer's instructions. The transfected cells were cultured in suspension for 8 days, and culture supernatants were collected and subjected to affinity chromatography using a HiTrap$^{TM}$ rProtein A FF column (Cytiva, USA, 17507901), as previously described [29]. Briefly, equilibration was performed using phosphate buffer solution (PB, pH 7.2), before loading the sample onto the equilibrated column. Following sample loading, the protein was eluted with 3.5 M sodium citrate–hydrochloric acid buffer solution (pH 2.7) and brought to a neutral pH with 1 M Tris base (pH 11.5). The concentration of purified IgG was determined using the Pierce BCA Protein Assay Kit (Thermo Fisher Scientific, USA, 23225). The sample was separated on a 4%–12% Bis-Tris gel (GenScript, USA, M00653) and stained with Commassie blue staining solution.

## ELISA

A 96-well high binding microplate was coated overnight at 4˚C with 100 µL per well of 2 µg/mL H1N1, H3N2, H5N1, and H7N9 NA recombinant proteins. After rinsing three times with PBST, the wells were blocked with 5% skimmed milk in PBST for 1 h at 37˚C. Serial dilutions of FNA1 mAb (seven dilutions), at a starting concentration of 5 µg/mL, were added and incubated at 37˚C for 1 h. After washing the plate three times, HRP-conjugated anti-human IgG (SeraCare, USA, 5220–0330) was added to each well. After incubating for 30 min and washing,

the peroxidase activity was detected by adding TMB substrate, and the reaction was stopped with 1 M $H_2SO_4$. Finally, the OD at 450 nm was recorded using a microplate reader.

## Western blot

For western blot, 1 μg of H1N1, H3N2, H5N1, and H7N9 NA recombinant proteins in reducing SDS loading buffer were heated in boiling water for 10 min and subjected to 4%–12% SDS PAGE. Separated proteins were transferred to a polyvinylidene fluoride (PVDF) blotting membrane (Merck, Germany, IPVH00010) using the eBlot L1 Fast Wet Transfer System (GenScript, USA). The blotted membrane was blocked with 5% skim milk in TBST (0.1% Tween 20 in Tris-buffered saline [TBS]) for 1 h and incubated with 5 μg/mL of FNA1 mAb in TBST containing 5% skimmed milk at 4°C overnight. After washing with TBST, the membrane was incubated with anti-human IgG secondary antibody for 1 h. After washing the membrane three times, the entire membrane was covered with an enhanced chemiluminescence substrate (PerkinElmer, USA, NEL105001EA), and images were acquired using a Chemiluminescence Imaging System (Clinx Science Instruments Co., China).

## Flow cytometry analysis

The CA/09 NA, A/Brisbane/59/2007 (H1N1, BR/07) NA, A/Puerto Rico/8/34 (H1N1, PR8) NA, and Anh05 NA genes were codon-optimized for human cells and cloned into the eukaryotic expression plasmid pcDNA3.1 to generate the recombinant plasmids. NA expression plasmids were transfected into 293T cells according to the instructions of Lipo3000 transfection reagent (Thermo Fisher Scientific, USA, L3000015). After 24 h, 293T cells were trypsinized, washed once with PBS, and incubated with 100 μL of 5 μg/mL FNA1 mAb for 30 min. After washing with PBS, the cells were incubated with fluorescein-labeled anti-human kappa light chain secondary antibody (Thermo Fisher Scientific, 12-9970-42) at 4°C for 30 min, washed twice with PBS, and detected by flow cytometry (BD, USA).

## Affinity determination

The affinity of the FNA1 mAb against CA/09 NA and Anh05 NA was quantitatively analyzed by bio-layer interferometry (BLI) using a ForteBio Octet System (Sartorius, Germany) according to the manufacturer's instructions. Briefly, CA/09 NA or Anh05 NA recombinant protein was diluted to 10 μg/mL with assay buffer (0.02% Tween 20 and 0.1% bovine serum albumin [BSA] in PBS), and the FNA1 mAb was serially diluted two-fold from 666 to 41.7 nM. Next, CA/09 NA or Anh05 NA protein was immobilized on the surface of NTA biosensors for 300 s. Then, the NA-captured biosensors were dipped into FNA1 mAb diluent for 300 s (association phase) and moved to assay buffer without analytes for 600 s (dissociation phase). Sensors were regenerated with three 5-s pulses of 10 mM glycine–HCl (pH 1.7). The data were analyzed using ForteBio data analysis software v. 9.0. The equilibrium dissociation constant (KD) indicates the affinity between the antibody and antigen.

## Production of H5N1 pseudoviruses

For pseudovirus construction, Anh05 HA and NA genes were codon-optimized for human cells and cloned into the eukaryotic expression plasmid pcDNA3.1 to generate the envelope recombinant plasmids pcDNA3.1-HA5 and pcDNA3.1-NA1. The packaging plasmid pNL4-3. Luc.R-E, containing the firefly luciferase gene, was gifted by the Institute of Microbiology, Chinese Academy of Sciences. Briefly, the pcDNA3.1-HA5, pcDNA3.1-NA1, and pNL4-3.Luc.R-E plasmids, at a mass ratio of 1:1:2, were transfected into 293T cells according to the instructions

of the Lipo3000 transfection reagent. Forty-eight hours after transfection, Anh05 pseudoviruses containing culture supernatants were harvested, passed through a 0.45-µm filter (Merck, Germany, SLHV033RB), and then concentrated by spinning at 4˚C 5000 g for 20 min using 30 KD centrifugal filters (Merck, Germany, UFC903096). For titration of the Anh05 pseudoviruses, serial ten-fold dilutions (nine dilutions in total) were made with DMEM medium and then used to infect MDCK cells in a 96-well culture plate at 100 µL/well. Wells without pseudovirus were designated as negative controls. After 48 h, the activities of firefly luciferases were measured on cell lysates using a luciferase substrate (Promega, USA, E1910) following the manufacturer's instructions. The positive well was determined to have relative luminescence unit (RLU) values that were three-fold higher than those of the control well. The 50% tissue culture infectious dose ($TCID_{50}$) was calculated using the Reed–Muench method, as described previously [30]. Subsequently, 20% FBS was added to the pseudoviruses before storing at –80˚C.

## NA enzyme-linked lectin assay (ELLA)

A high-binding 96-well plate was coated with 5 µg/mL fetuin (Sigma, Germany, SRP3283) at 4˚C overnight. The concentrated Anh05 pseudoviruses were diluted 20-fold in the sample diluent (DPBS; Gibco, 14190–144) with 1% BSA and 0.5% Tween 20). The FNA1 mAb or WN1 (an anti-West Nile virus antibody prepared by our group) was serially diluted (five-fold; seven dilutions in total) with the diluted pseudovirus solution at a starting concentration of 100 µg/mL and incubated at 37˚C for 1 h. After washing with wash buffer (0.05% Tween 20 in PBS) three times, mixtures of pseudoviruses and antibody were added into the fetuin plate, at 100 µL/well, and incubated for 18 h at 37˚C, buffered by $CO_2$ as for tissue culture. On the next day, the plate was washed six times with wash buffer, and HRP-conjugated peanut agglutinin lectin (PNA-HRP; Sigma, L7759; 2.5 µg/mL) was added to the wells. After a 2-h incubation and washing with wash buffer, the signal was developed by adding TMB solution, and 1M $H_2SO_4$ was used to stop the reaction. The absorbance was read at 450 nm using a microplate reader. The inhibition of CA/09 NA (diluted to 25 µg/mL with the sample diluent) activity or Anh05 NA (diluted to 0.5 µg/mL) activity by FNA1 mAb were measured using ELLA as described above.

## NA-XTD assay

The NA-XTD assay was performed as described in the instructions of the NA-XTD™ Influenza Neuraminidase Assay Kit (Thermo Fisher Scientific, USA, 4457535). Briefly, 25 µL of FNA1 mAb in serial two-fold dilutions in NA-XTD assay buffer was mixed with 25 µL of the concentrated Anh05 pseudoviruses diluted 20-fold with the assay buffer. After incubation at 37˚C for 1 h, 25 µL of NA-XTD substrate was added and incubated at room temperature for 30 min. The reaction was stopped by adding 60 µL of NA-XTD Accelerator. Chemiluminescence was determined by using a microplate luminometer (Molecular Devices, USA).

## In vitro inhibition of pseudovirus release by FNA1 mAb

PcDNA3.1-HA5, pcDNA3.1-NA1, and pNL4-3.Luc.R-E plasmids at a mass ratio of 1:1:2 were transfected into 293T cells according to the instructions of the Lipo3000 transfection reagent. Six hours after transfection, fresh DMEM medium containing 5 µg/mL or 50 µg/mL FNA1 mAb was added to the corresponding wells, and wells without antibody were set as controls. After 48 h of culture, cell supernatants from different treatment wells were collected and passed through 0.45-µm filter units, then used to infect 293T cells in a 24-well plate. After 48 h of infection, the activities of firefly luciferases were measured on cell lysates using a luciferase

substrate following the manufacturer's instructions. Assessments of inhibition of PR8 and BR/07 pseudovirus release by FNA1 mAb were performed similarly as described above.

### Neutralization test of the FNA1 mAb in vitro

The concentrated Anh05 pseudoviruses were diluted 100-fold in DMEM medium, and FNA1 mAb or FHA3 (an anti-HA antibody prepared by our group) was diluted to 5 μg/mL or 50 μg/mL with the diluted pseudovirus solution, and incubated at 37°C for 1 h. The pseudovirus solution without antibodies was used for the negative controls. Subsequently, the mixture was inoculated into 80% confluent MDCK cells in a 96-well plate. After a 6 h incubation, DMEM with 10% FBS was added to the cell wells and cultured for 2 days. The activities of firefly luciferases were measured on infected MDCK cell lysates using a luciferase substrate following the manufacturer's instructions.

### Preliminary analysis of the antigenic site recognized by FNA1

After consulting the literature, 35 mutation sites reported to affect the binding of antibodies and NA proteins were selected. Subsequently, primers for these mutants were designed and synthesized by Sangon Biotech. Site-directed mutagenesis was performed on the H1N1 NA gene from the A/California/04/2009 strain by PCR according to the instructions of the Fast Site-Directed Mutagenesis Kit (TIANGEN, China, KM101). These mutant genes were cloned into the pcDNA3.1 plasmid to construct the NA mutant plasmids. After transfection of the plasmids into 293T cells, the binding activity of FNA1 mAb and NA mutants was detected by flow cytometry, as described previously.

### Data analysis

Graphs were generated using GraphPad Prism software v.8.4.3. The ELLA titer was expressed as the half-maximal inhibitory concentrations ($IC_{50}$) determined using a sigmoidal dose–response curve. One-tailed unpaired Student's t-test was used to evaluate significant differences. Values were considered statistically significant when $p < 0.05$. The results of flow cytometry were analyzed using FlowJo software.

## Results

### Selection and preparation of antibodies targeting N1 NAs

A synthetic scFv phage display library with an estimated diversity of $3 \times 10^6$ colony forming units (CFUs) was used to select novel antibodies that recognize H1N1 NA. After the third round of biopanning, 88 individual clones were randomly selected from the plate to prepare monoclonal phages, and the binding ability of monoclonal phages to CA/09 NA was determined by ELISA. Among the 88 clones, 33 were considered to be positive ($OD_{450} > 0.5$, Fig 1A), and the DNA encoding positive scFv clones was sequenced. One scFv, designated FNA1, was focused on and selected for the presentation of results due to its good reactivity against N1 NA antigens (Fig 2).

To produce the binder as an antibody protein, an FNA1 mAb expression plasmid was constructed by inserting the heavy and light chain variable region sequences into the mammalian expression vector pFRT-IgG1κ. After transfecting the construct, transient expression in Expi-CHO cells resulted in antibody molecules, which were then purified from the culture supernatant by affinity chromatography on a protein A column. The purity and integrity of the purified antibody proteins were confirmed by SDS-PAGE analysis, which revealed a purity of more than 95% (Fig 1B).

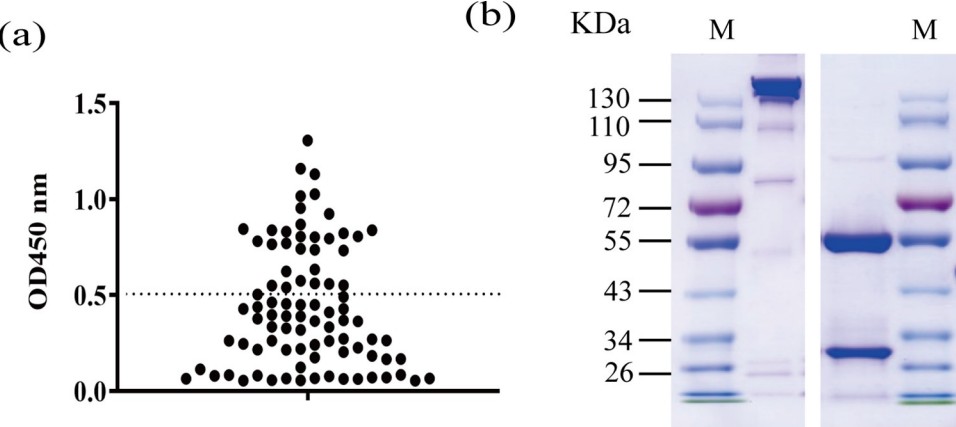

**Fig 1. Isolation of an IgG-based monoclonal antibody (mAb) targeting H1N1 NA protein.** (a) Following three rounds of panning on phages binding immobilized H1N1 NA, single clones were selected and superinfected for phage production, and the produced phages were tested for H1N1 NA binding by ELISA. Each point represents the $OD_{450}$ value measured for a monoclonal phage. (b) SDS-PAGE analysis under non-reducing (left) and reducing (right) conditions for FNA1 mAb.

## Validation of the antigen binding of FNA1 mAb

The binding capacity of the obtained FNA1 mAb to NA proteins was first tested by ELISA. The results showed that FNA1 could bind to not only CA/09 NA but also Anh05 NA in a concentration-dependent manner but had no cross interaction with the NAs of H3N2 and H7N9, suggesting that FNA1 mAb specifically recognizes N1 NAs (Fig 2A). We next verified the reactivity of FNA1 to CA/09 NA and Anh05 NA by western blot, the results of which were consistent with those of the ELISA (Fig 2B). Additionally, we found that FNA1 had limited binding breadth, and, as shown in Fig 2C, did not react to PR8 NA or BR/07 NA (Fig 2C). Through aligning the NA amino acid sequences of the six strains of influenza viruses discussed above, we found that Anh05 NA showed a higher homology with CA09 NA than the other sequences (S1 Fig and S1 Table). Next, we conducted binding kinetic analyses based on BLI measurements to determine the FNA1 mAb affinity to CA/09 NA or Anh05 NA. The results showed that FNA1 bound to CA/09 NA and Anh05 NA with equilibrium KD values of $7.82 \times 10^8$ mol/L and $2.97 \times 10^8$ mol/L, respectively (Fig 2D).

## FNA1 mAb inhibits NA enzymatic activity and Anh05 pseudovirus release in vitro

Next, to determine whether FNA1 is capable of inhibiting NA activity in vitro, we conducted an ELLA and NA-XTD assay. Because the packaged CA/09 pseudoviruses are not sufficiently stable and their infectivity is low, only Anh05 pseudoviruses were used to assess the in vitro function of FNA1. A batch of Anh05 pseudoviruses was prepared and the pseudovirus titer was measured. In these two experiments, the titer of pseudoviruses used was $2.59 \times 10^5$ $TCID_{50}$/mL after dilution. In the ELLA assay, FNA1 decreased the cleavage of the large-molecule substrate fetuin by Anh05 pseudoviruses by 50% at a low concentration ($IC_{50}$ of 0.97 μg/mL, Fig 3A), but failed to inhibit the cleavage of the small-molecule substrate NA-XTD by pseudovirus NAs (S2 Table). Furthermore, recombinant CA/09 NA and Anh05 NA were used in ELLA to evaluate the NI activity of FNA1 mAb. We found that FNA1 showed a significant inhibitory effect on both CA/09 NA and Anh05 NA, with $IC_{50}$ values of 0.06 μg/mL and 2.88 μg/mL, respectively (Fig 3B). These results suggest that the NI activity of the antibody may be not caused by direct binding to or an allosteric effect on the enzyme active site, but

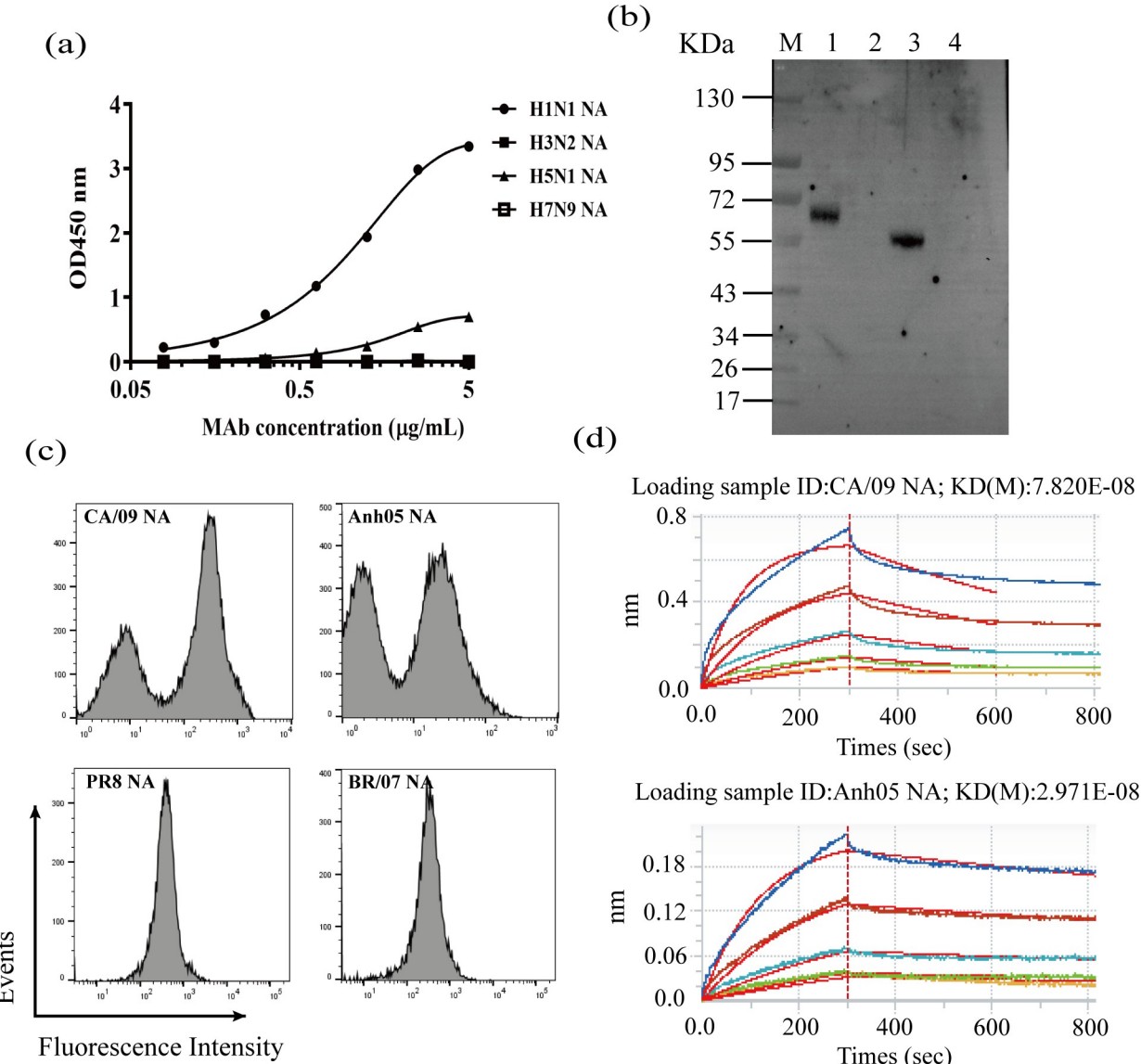

**Fig 2. Reactivity of FNA1 mAb to N1 NAs.** (a) The antigen-binding activities of purified FNA1 mAb to the NAs of H1N1, H3N2, H5N1, and H7N9 were assessed by ELISA. (b) Western blot analysis of FNA1 mAb. M: Protein marker, lane 1: H1N1 NA (His Tag, 58.7 kDa), lane 2: H3N2 NA (His Tag, 50.5 kDa), lane 3: H5N1 NA (His Tag, 55 kDa), lane 4: H7N9 NA (His Tag, 61–69 kDa). (c) The binding of FNA1 to CA/09 NA, Anh05 NA, PR8 NA, and BR/07 NA expressed transiently on the cell membrane of 293T cells was demonstrated by flow cytometry. (d) The binding affinity of FNA1 mAb to CA/09 NA or Anh05 NA was measured by the ForteBio method. Each BLI sensorgram shows the interaction between different concentrations of FNA1 and the CA/09 NA- or Anh05 NA-immobilized sensor chip.

because FNA1 binds to the region near the active site and blocks the access of the enzymatic site to sialic acid through steric hindrance.

We also explored whether the FNA1 mAb could inhibit pseudovirus spread. During the final stages of viral replication, NA plays a major role in releasing viral progeny; thus, anti-NA mAbs are mostly effective during viral egress [14]. In our assay, we co-transfected the pNL4-3.Luc.R-E, pcDNA3.1-HA5, and pcDNA3.1-NA1 plasmids into 293T cells to package Anh05 pseudovirus particles. Six hours after transfection, fresh medium containing FNA1 mAb was replaced to block pseudovirus release. The results show that the RLUs measured in the FNA1 groups were significantly reduced compared to those in the control group (Fig 3C), indicating that FNA1

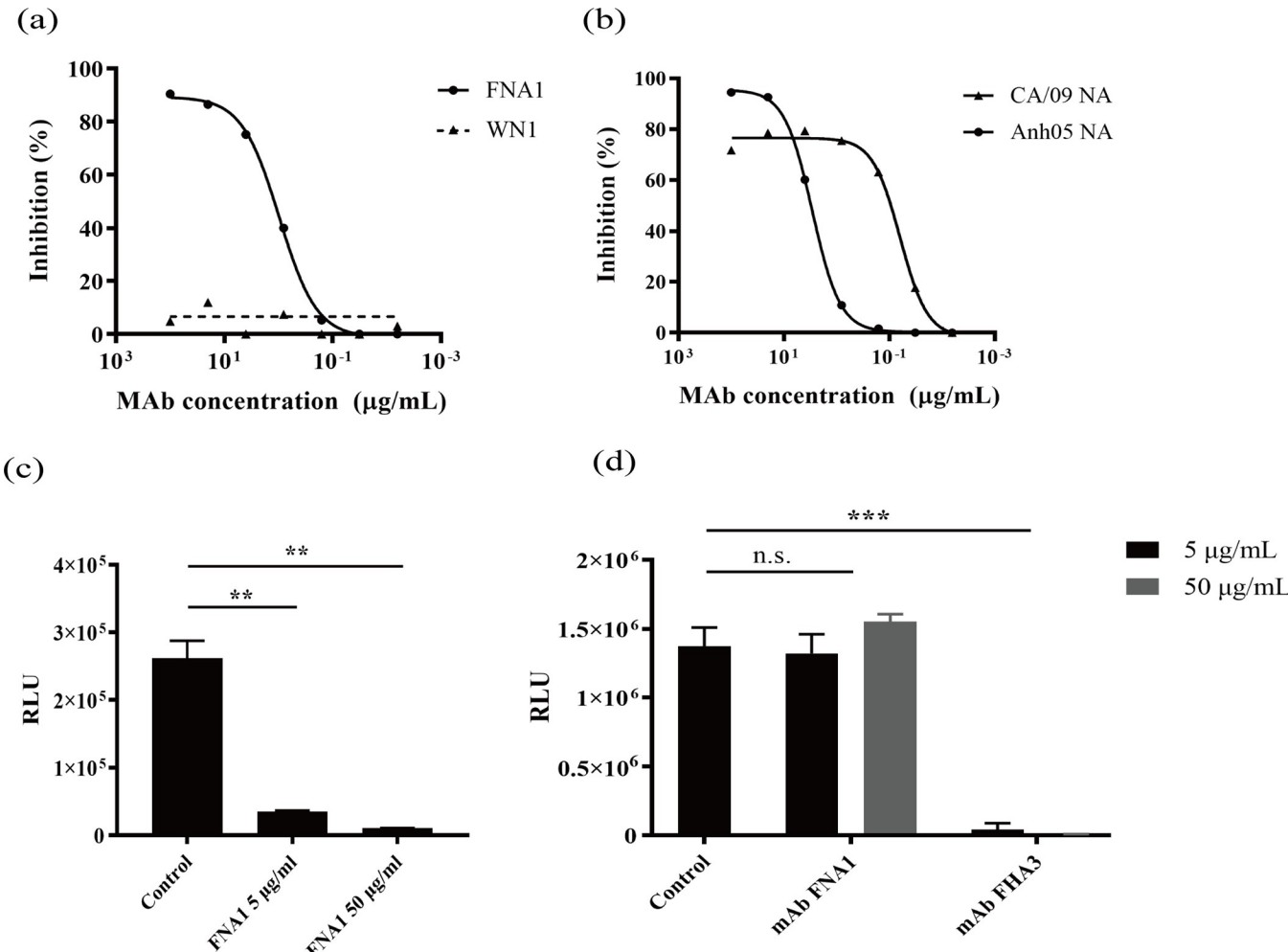

**Fig 3. Inhibitory effect of FNA1 mAb against N1 NAs and pseudovirus release.** (a) NI activity of FNA1 mAb against H5N1 pseudoviruses measured via ELLA, with WN1 mAb used as a negative control. (b) Testing of FNA1 mAb's inhibition of CA/09 NA and Anh05 NA enzymatic activity in ELLA assays. (c) Inhibition of H5N1 pseudovirion egress by FNA1 mAb. (d) Determination of the neutralization activity of FNA1 mAb against Anh05 pseudoviruses, with the anti-influenza virus neutralizing mAb FHA3 serving as a positive control.

effectively blocked the shedding of newly packaged pseudovirus particles from the cell surface, thereby inhibiting further infection of cells. However, FNA1 could not inhibit the spread of PR8 and BR/07 pseudoviruses (S3 Table), which agrees well with the results of the binding assay.

Although in theory anti-NA mAbs cannot prevent viral infection, they may interrupt the infection by interfering with neighboring HA molecules because of the binding of NA proteins. Therefore, we also tested whether the bound FNA1 could hinder pseudovirus infection. In the neutralization test, FNA1 did not prevent H5N1 pseudovirus infection of MDCK cells as was observed with anti-HA antibodies (Fig 3D), demonstrating that FNA1 does not control virus infection by blocking virus attachment and entry into cells.

## Preliminary epitope identification by detection of binding of FNA1 mAb and NA mutants

According to multiple reports in the literature, 35 amino acid residues of NA protein that may affect antibody binding were selected for site-directed mutagenesis, and NA mutant expression

plasmids were constructed. The plasmids were transfected into 293T cells, and flow cytometry was used to test the reactivity of FNA1 to H1N1 NA mutants. The results showed that FNA1 could not bind to mutants 15, 21, 31, and 33 (Fig 4), suggesting that W219, K254, W358, and S388 were the key residues that affected the recognition of H1N1 NA by FNA1.

## Discussion

N1 subtype influenza A viruses, such as H1N1 and H5N1, are common respiratory pathogens. H1N1 viruses are widespread and represent the main causative agents of seasonal influenza outbreaks worldwide [31, 32]. H5N1 avian influenza viruses can cause severe infection in humans, with a mortality rate of approximately 60% [33, 34]. Therefore, it is becoming increasingly important to seek effective therapeutics and prophylactics to control viral infections.

The enzyme activity site of NA has long been the focus of antiviral drug designs [35]. Therapeutic antibodies targeting NA are constantly being developed, and most of them have broadly cross-reactive activity. Stadlbauer et al. [27] described a set of mAbs, including 1G04, 1E01, and 1G01, which show broad heterosubtypic binding to NAs from influenza A group 1 (N1, N4, N5, N8) and 2 (N2, N3, N6, N7, N9), and influenza B viruses, and are potent inhibitors of NA activity. DA03E17 [28] and HCA-2 [36] are analogous to 1G04, 1E01, and 1G01, with consistent binding breadth. N1-C4 [37], reported by Job E R et al., and multiple antibodies [17], reported by Strohmeier et al., are some of the subtype-specific antibodies that target N1 and N6 NAs, respectively. These antibodies can also mediate NI and reduce virus replication in vitro. Here, we screened a strain of FNA1 mAb of N1 NAs using the phage display approach. FNA1 binds to H1N1 and H5N1 NAs but does not react with H3N2 and H7N9 NAs. Nonetheless, FNA1 is not broadly reactive and can recognize CA/09 NA and Anh05 NA but not PR8 NA and BR/07 NA. We did not test more NA subtypes because CA/09 NA was used as the target antigen during the phage display selections; as a result, the obtained antibodies were probably N1-specific. Additionally, by aligning the NA amino acid sequences of the influenza viruses described in the study, we found that Anh05 NA showed a higher homology between Anh05 NA and CA/09 NA compared to the other sequences. In the site-directed mutagenesis experiment, we found that the substitution of W219A, K254A, W358A, and S388A led to a loss of FNA1 binding activity on H1N1 NA, revealing that these NA residues were likely key contacts for antibody binding.

In our study, a pseudovirus system was used to assess the in vitro functionality of FNA1 mAb. Pseudoviruses are a type of recombinant viral particle whose backbone and envelope proteins are derived from different viruses [38]. Genes of the viral vector are typically modified so that they are unable to produce their own surface protein, which is then replaced with the surface glycoprotein of the designated virus [39, 40]. The producing pseudoviruses can infect host cells but replicate intracellularly for only a single cycle [41]. Although their amplification is defective, pseudoviruses have been widely used in virus-related research because of their strong operability and low biological risk [42]. Our experiment revealed that FNA1 inhibited the sialidase activity of not only the CA/09 NA and Anh05 NA proteins, but also H5N1 pseudoviruses, by blocking access of the substrate to the enzyme-active site. This inhibition was observed in the cleavage of large-molecule fetuin, but not the small substrate NA-XTD. FNA1 also demonstrated the ability to block the release of virus particles from packaging cells, thus inhibiting pseudovirus infection. These findings indicate the presence of in vitro protection by FNA1 and suggest its potential protective role in vivo.

Many NI antibodies, including AG7C, AF9C, Z2B3 [43], 3c10-3 [24], and 1G8 [44], have been found to protect mice against lethal infections with IAVs and mitigate the severity of

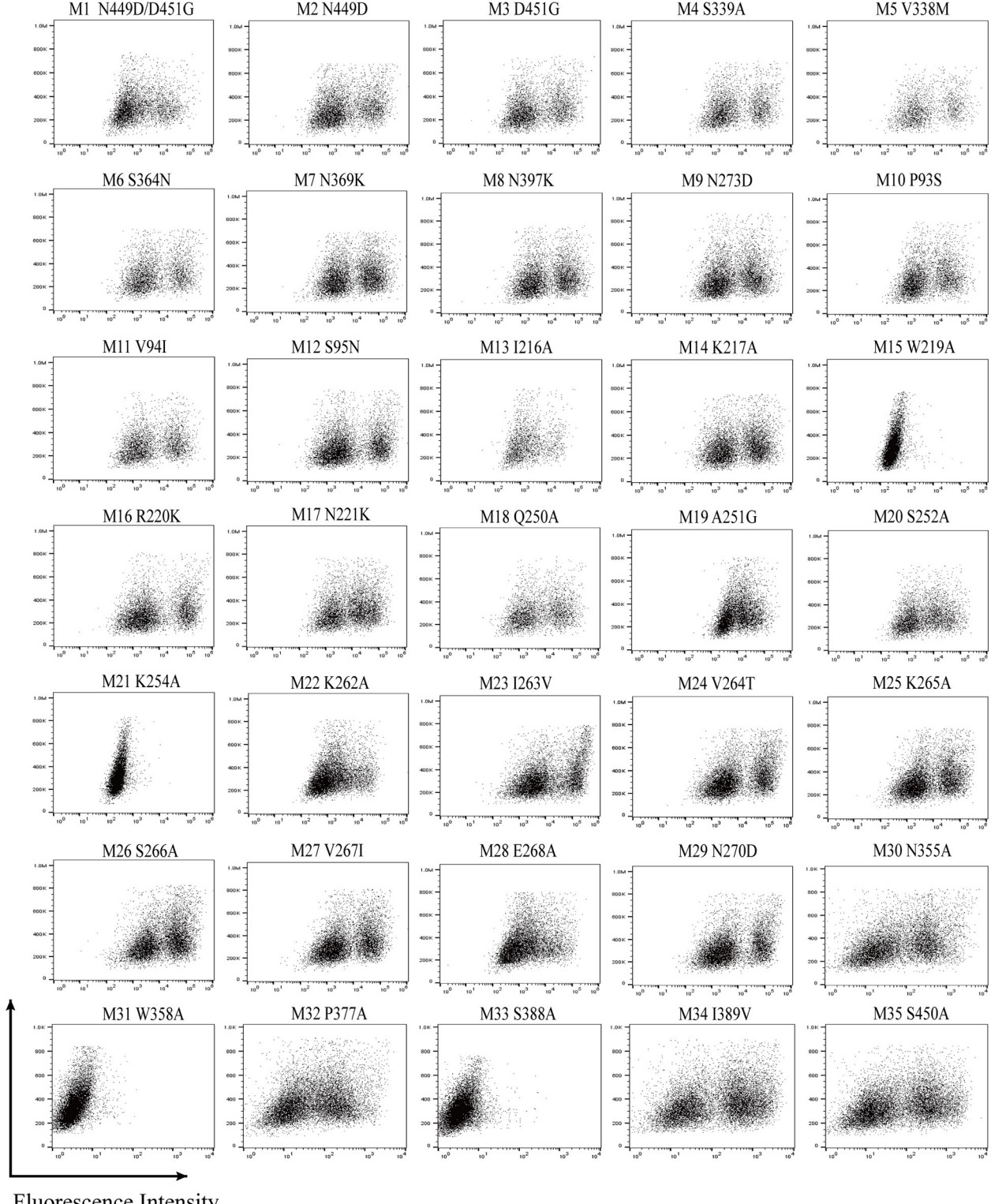

**Fig 4. Identification of the effect of NA point mutations on the binding of FNA1 mAb to H1N1 NA involved the transient expression of NA mutants on the surface of 293T cells.** The binding activities of FNA1 mAb and the mutants were then detected by flow cytometry.

disease. Similarly, we performed animal experiments to evaluate the protective effects of FNA1 mAb against H5N1 pseudovirus infection. However, FNA1 was not found to reduce pseudovirus infection; this may be because anti-NA mAbs exert their function at the stage when the influenza virions bud off from the infected cells rather than preventing viral infection, and pseudoviruses cannot simulate the process of proliferation and release after entry into cells [42]. As a consequence, pseudoviruses could still infect mice and bioluminescence light could be detected. Furthermore, we also utilized the live influenza viruses in a mouse challenge model. Regrettably, we currently still lack access to the CA/09 virus. We carried out an in vivo prophylactic study to evaluate FNA1 mAb's potential in protecting against PR8. Consistent with the binding assay result, FNA1 failed to protect mice from PR8-induced lethality and weight loss, as shown in S2 Fig. In this regard, further experiments still need to be conducted to better understand the function of FNA1 mAb in protecting against H1N1 and H5N1 influenza virus challenge.

In summary, we have prepared and characterized a mAb, FNA1, which specifically recognizes the NA of N1 subtype influenza viruses. Through the analysis of the antiviral properties in vitro, we found that FNA1 could effectively inhibit N1 NA activity and pseudovirus spread, which is a prerequisite for its therapeutic role in vivo. These results suggest that FNA1 has potential value as a drug for influenza treatment.

## Supporting information

**S1 File. Supporting data for the experiments shown in the figures, excluding flow cytometry experiments.**
(DOCX)

**S1 Fig. Phylogenetic tree analysis of the NAs of influenza viruses described in the present study.**
(PNG)

**S2 Fig. Determination of the protective activity of mAb FNA1 against PR8 lethality.**
(TIF)

**S1 Table. Alignment of influenza virus NA amino acid sequences described in the present study.**
(XLSX)

**S2 Table. NI activity of FNA1 mAb against Anh05 pseudoviruses measured by the NA-XTD assay.** Anti-West Nile virus mAb WN1 was used as a negative control.
(XLSX)

**S3 Table. Detection of in vitro inhibition of PR8 or BR07 pseudovirus release by FNA1.**
(XLSX)

**S4 Table. Raw data for the mouse PR8 lethality challenge experiment.**
(XLSX)

## Author Contributions

**Data curation:** Nan Chen.

**Formal analysis:** Nan Chen, He Xiao.

**Methodology:** Guojiang Chen, Chenghua Liu, He Xiao.

**Project administration:** Beifen Shen, Jiannan Feng.

**Supervision:** Jing Wang, Lihui Chai, Zuyin Yu, He Xiao.

**Validation:** Nan Chen, Wanlu Zhu, Xiangjun Hao, He Xiao.

**Writing – original draft:** Nan Chen.

**Writing – review & editing:** Renxi Wang, ChunXia Qiao, Xinying Li, Lihui Chai, Zuyin Yu, He Xiao.

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
