## [Decision Letter · Decision Letter 0]

22 Feb 2024

PONE-D-23-41925Development and characterization of an antibody recognizing the influenza virus N1 neuraminidasesPLOS ONE

Dear Dr. Xiao,

Thank you for submitting your manuscript to PLOS ONE. After careful consideration, we feel that it has merit but does not fully meet PLOS ONE’s publication criteria as it currently stands. Therefore, we invite you to submit a revised version of the manuscript that addresses the points raised during the review process. During the revision process, please address comments related to release and presentation of sequence data as well as the comments and/or experiments associated with a challenge study that demonstrates protection against infection.

We look forward to receiving your revised manuscript.

Kind regards,

Victor C Huber

Academic Editor

PLOS ONE

Reviewers' comments:

Reviewer's Responses to Questions

**Comments to the Author**

1. Is the manuscript technically sound, and do the data support the conclusions?

Reviewer #1: Partly

Reviewer #2: Yes

2. Has the statistical analysis been performed appropriately and rigorously? 

Reviewer #1: Yes

Reviewer #2: Yes

3. Have the authors made all data underlying the findings in their manuscript fully available?

Reviewer #1: Yes

Reviewer #2: Yes

4. Is the manuscript presented in an intelligible fashion and written in standard English?

Reviewer #1: Yes

Reviewer #2: Yes

5. Review Comments to the Author

Reviewer #1: This is a straightforward study on the isolation and characterization of human mAb against neuraminidase. Specifically, the mAb appears to be directly targeting N1 as shown in ELISA, cell expression by flow cytometry, and W.B. Furthermore, neuraminidase inhibition assay was performed with finding suggesting rather potent inhibition of enzymatic activities of NA. Moreover, it seems that a.a. residues 219, 254, 358, and 388 in the NA protein were involved in the binding of the viral protien to the mAb.

In general, this is a well designed experimental study with data rather clearly being presented. The following points are put forth for the authors to consider:

1. Did the authors test NA subtypes beyond N1, N2 and N7? if not, please explain. Also, a sequence line-up of the NA sequences of the NA used in this study would be helpful for the readerships.

2. While this reviewer does like the work, the manuscript, as it stands now, is quite incomplete in terms of being convincingly showcasing the power of the mAb. Clearly, this work could really be elevated to a much higher visibility by conducting protection experiment in mouse model where mAb is shown to demonstrate protection against chalenge by H1N1, not H3N2 etc.

Reviewer #2: In this manuscript, the author got a human cross-reactive H1N1 and H5N1 N1 antibody FNA1 by using phage display. Moreover, this antibody showed NA inhibition and blocking of the virus release and this may be due to steric hinderance. The story is straight-forward and I only have minor comments.

1. For the antibody screened from phage display, is that only one antibody have binding to N1 or have more?

2. The author should submit the FNA1 antibody sequence to public database, like NCBI.

3. For suggestion, the author can use NA protein for ELLA experiments. So the author can test ELLA activity against CA/09 N1.

4. In Figure 3c, the author should describe what is FHA3.

In this manuscript, the author obtained a human cross-reactive H1N1 and H5N1 N1 antibody, FNA1, using phage display. Furthermore, this antibody demonstrated NA inhibition and the blocking of virus release, potentially attributable to steric hindrance. The story is straightforward and I only have minor comments.

1. Is the antibody screened from phage display the only one binding to N1, or are there others?

2. The author should consider submitting the FNA1 antibody sequence to public databases like NCBI.

3. As a suggestion, the author could use NA protein for ELLA experiments to assess ELLA activity against CA/09 N1.

4. In Figure 3c, the author should clarify the nature of FHA3.

6. PLOS authors have the option to publish the peer review history of their article (what does this mean?). If published, this will include your full peer review and any attached files.

Reviewer #1: No

Reviewer #2: No

---

## [Author Response · Author response to Decision Letter 0]

27 Mar 2024

Response to academic editor

 Response: In accordance with PLOS ONE's style requirements, we have adjusted the formatting of our manuscript entitled “Development and characterization of an antibody recognizing the influenza virus N1 neuraminidases” and followed the file naming process.

Response: ORCID iDs of the three corresponding authors of this manuscript have been successfully linked to their Editorial Manager accounts following the provided instructions.

Response: We acknowledge this suggestion and have added tables in the supporting information (S2 and S3 Tables) to confirm our statement.

Response: We have updated the supporting information files by replacing the original ones with a Word file (S1_File.docx), a figure (S1_Fig.png), and three Excel files (S1_Table.xlsx, S2_Table.xlsx, and S3_Table.xlsx). In addition, captions for the supporting information files have been added at the end of our manuscript, following the guidelines for supporting information. We have also updated the in-text citations accordingly (line 292, line 306, and line 330).

Response to Reviewer #1

1. Did the authors test NA subtypes beyond N1, N2 and N7? if not, please explain. Also, a sequence line-up of the NA sequences of the NA used in this study would be helpful for the readerships.

Response: We focused our testing exclusively on NA subtypes N1 (CA/09 NA, PR8 NA, BR/07 NA, and Anh05 NA), N2 (H3N2 NA), and N9 (H7N9 NA). The reason for this is that CA/09 NA was used as the target antigen during phage display selections. Therefore, the antibodies obtained are probably N1-specific. While FNA1 demonstrated binding to CA/09 and Anh05 NA, it exhibited no reactivity with PR8, BR/07, H3N2, and H7N9 NA, which shows its limited binding breadth. Additionally, in response to the reviewer’s recommendation, we aligned the NA amino acid sequences of influenza viruses described in the study, and found that Anh05 NA showed a higher homology between Anh05 NA and CA/09 NA compared to other sequences (line 290–292, S1 Fig and S1 Table).

2. While this reviewer does like the work, the manuscript, as it stands now, is quite incomplete in terms of being convincingly showcasing the power of the mAb. Clearly, this work could really be elevated to a much higher visibility by conducting protection experiment in mouse model where mAb is shown to demonstrate protection against challenge by H1N1, not H3N2 etc.

Response: We acknowledge the reviewer’s suggestion regarding mouse challenge experiments in terms of demonstrating the protective capacity of the mAb against H1N1, but not other subtypes, e.g., H3N2, which could significantly improve the visibility of our study. Regrettably, we currently lack access to the CA/09 virus. Nevertheless, we carried out an in vivo prophylactic study to evaluate FNA1 mAb's potential in protecting against PR8. Despite our efforts, FNA1 failed to provide protection against PR8, as shown in the subsequent figure. 

Response to Reviewer #2

1. Is the antibody screened from phage display the only one binding to N1, or are there others?

Response: Through the use of phage display techniques, we identified three mAbs targeting N1 NAs. While all three mAbs showed binding activity to CA/09 and Anh05 NA, the affinities (KD) of the two other mAbs were lower compared to FNA1. In addition, analysis revealed that these three mAbs shared the same four key residues of H1N1 NA that affected their binding to viral proteins, suggesting that they might target similar sites. Based on these findings, we opted to focus solely on FNA1 for presenting the results.

2. The author should consider submitting the FNA1 antibody sequence to public databases like NCBI.

Response: We concur with the reviewer’s suggestion regarding the submission of the FNA1 mAb sequence to a public database. Consequently, we have deposited the DNA sequences of both the FNA1 heavy and light chain variable regions at GenBank. The accession numbers for the nucleotide sequences are PP483115 (heavy chain) and PP483116 (light chain).

3. As a suggestion, the author could use NA protein for ELLA experiments to assess ELLA activity against CA/09 N1.

Response: We agree with the reviewer’s suggestion to utilize NA protein in ELLA experiments to assess the NI activity of FNA1 mAb against CA/09 N1. Accordingly, we conducted ELLA assays using recombinant CA/09 NA and Anh05 NA, and found that FNA1 effectively blocked the activity of both CA/09 NA and Anh05 NA, resulting in the decreased cleavage of fetuin by NA proteins, with IC50 values of 0.06 μg/mL and 2.88 μg/mL, respectively (line 306–309 and updated Fig 3b).

4. In Figure 3c, the author should clarify the nature of FHA3.

Response: We greatly appreciate the reviewer's suggestion and have added a clarifying remark in line 319.

---

## [Decision Letter · Decision Letter 1]

4 Apr 2024

PONE-D-23-41925R1Development and characterization of an antibody that recognizes influenza virus N1 neuraminidasesPLOS ONE

Dear Dr. Xiao,

Thank you for submitting your manuscript to PLOS ONE. After careful consideration, we feel that it has merit but does not fully meet PLOS ONE’s publication criteria as it currently stands. Therefore, we invite you to submit a revised version of the manuscript that addresses the points raised during the review process. During the revision process, please clearly state and expand on the limitations of the study.

We look forward to receiving your revised manuscript.

Kind regards,

Victor C Huber

Academic Editor

PLOS ONE

Journal Requirements:

Reviewers' comments:

Reviewer's Responses to Questions

**Comments to the Author**

1. If the authors have adequately addressed your comments raised in a previous round of review and you feel that this manuscript is now acceptable for publication, you may indicate that here to bypass the “Comments to the Author” section, enter your conflict of interest statement in the “Confidential to Editor” section, and submit your "Accept" recommendation.

Reviewer #1: (No Response)

2. Is the manuscript technically sound, and do the data support the conclusions?

Reviewer #1: Partly

3. Has the statistical analysis been performed appropriately and rigorously? 

Reviewer #1: I Don't Know

4. Have the authors made all data underlying the findings in their manuscript fully available?

Reviewer #1: Yes

5. Is the manuscript presented in an intelligible fashion and written in standard English?

Reviewer #1: Yes

6. Review Comments to the Author

Reviewer #1: Please see my responses to your revision Ms:

Response to Reviewer #1

1. Did the authors test NA subtypes beyond N1, N2 and N7? if not, please explain. Also, a sequence line-up of the NA sequences of the NA used in this study would be helpful for the readerships.

Response: We focused our testing exclusively on NA subtypes N1 (CA/09 NA, PR8 NA, BR/07 NA, and Anh05 NA), N2 (H3N2 NA), and N9 (H7N9 NA). The reason for this is that CA/09 NA was used as the target antigen during phage display selections. Therefore, the antibodies obtained are probably N1-specific. While FNA1 demonstrated binding to CA/09 and Anh05 NA, it exhibited no reactivity with PR8, BR/07, H3N2, and H7N9 NA, which shows its limited binding breadth. Additionally, in response to the reviewer’s recommendation, we aligned the NA amino acid sequences of influenza viruses described in the study, and found that Anh05 NA showed a higher homology between Anh05 NA and CA/09 NA compared to other sequences (line 290–292, S1 Fig and S1 Table).

Reviewer's responses to the latest revised Ms: Understood your limitations. Yet, such limitations should be more elaborated in the Discussion.

2. While this reviewer does like the work, the manuscript, as it stands now, is quite incomplete in terms of being convincingly showcasing the power of the mAb. Clearly, this work could really be elevated to a much higher visibility by conducting protection experiment in mouse model where mAb is shown to demonstrate protection against challenge by H1N1, not H3N2 etc.

Response: We acknowledge the reviewer’s suggestion regarding mouse challenge experiments in terms of demonstrating the protective capacity of the mAb against H1N1, but not other subtypes, e.g., H3N2, which could significantly improve the visibility of our study. Regrettably, we currently lack access to the CA/09 virus. Nevertheless, we carried out an in vivo prophylactic study to evaluate FNA1 mAb's potential in protecting against PR8. Despite our efforts, FNA1 failed to provide protection against PR8, as shown in the subsequent figure.

Reviewer's responses to the latest revised Ms: Understood your limitations. Yet, such limitations should be more clearly elaborated in the Discussion.

7. PLOS authors have the option to publish the peer review history of their article (what does this mean?). If published, this will include your full peer review and any attached files.

Reviewer #1: No

---

## [Author Response · Author response to Decision Letter 1]

11 Apr 2024

Article Title: Development and characterization of an antibody recognizing the influenza virus N1 neuraminidases

Running Number: PONE-D-23-41925R1

Response to academic editor

Response: We acknowledge this suggestion and have reviewed the reference list carefully. We discovered that the 30th reference was incomplete, as it lacked the identifier (DOI), and the 31st reference was a manuscript that had just been accepted. Consequently, these two references were removed and replaced with relevant current references (line 499–504).

Response to Reviewer #1

1. Did the authors test NA subtypes beyond N1, N2 and N7? if not, please explain. Also, a sequence line-up of the NA sequences of the NA used in this study would be helpful for the readerships.

Response: We focused our testing exclusively on NA subtypes N1 (CA/09 NA, PR8 NA, BR/07 NA, and Anh05 NA), N2 (H3N2 NA), and N9 (H7N9 NA). The reason for this is that CA/09 NA was used as the target antigen during phage display selections. Therefore, the antibodies obtained are probably N1-specific. While FNA1 demonstrated binding to CA/09 and Anh05 NA, it exhibited no reactivity with PR8, BR/07, H3N2, and H7N9 NA, which shows its limited binding breadth. Additionally, in response to the reviewer’s recommendation, we aligned the NA amino acid sequences of influenza viruses described in the study, and found that Anh05 NA showed a higher homology between Anh05 NA and CA/09 NA compared to other sequences (line 290–292, S1 Fig and S1 Table).

Reviewer's responses to the latest revised Ms: Understood your limitations. Yet, such limitations should be more elaborated in the Discussion.

Response: We concur with the reviewer’s suggestion regarding the exposition of the limitations of our study in the Discussion, and we have added further relevant content to clarify the limitations (line 370–374).

2. While this reviewer does like the work, the manuscript, as it stands now, is quite incomplete in terms of being convincingly showcasing the power of the mAb. Clearly, this work could really be elevated to a much higher visibility by conducting protection experiment in mouse model where mAb is shown to demonstrate protection against challenge by H1N1, not H3N2 etc.

Response: We acknowledge the reviewer’s suggestion regarding mouse challenge experiments in terms of demonstrating the protective capacity of the mAb against H1N1, but not other subtypes, e.g., H3N2, which could significantly improve the visibility of our study. Regrettably, we currently lack access to the CA/09 virus. Nevertheless, we carried out an in vivo prophylactic study to evaluate FNA1 mAb's potential in protecting against PR8. Despite our efforts, FNA1 failed to provide protection against PR8, as shown in the subsequent figure.

Reviewer's responses to the latest revised Ms: Understood your limitations. Yet, such limitations should be more clearly elaborated in the Discussion.

Response: We greatly appreciate the reviewer's suggestion and have clearly stated the limitations in the Discussion (line 400–404).

---

## [Editor Report · Decision Letter 2]

15 Apr 2024

Development and characterization of an antibody that recognizes influenza virus N1 neuraminidases

PONE-D-23-41925R2

Dear Dr. Xiao,

We’re pleased to inform you that your manuscript has been judged scientifically suitable for publication and will be formally accepted for publication once it meets all outstanding technical requirements.

Kind regards,

Victor C Huber

Academic Editor

PLOS ONE
---

## [Editor Report · Acceptance letter]

29 Apr 2024

PONE-D-23-41925R2 

PLOS ONE

Dear Dr. Xiao, 

I'm pleased to inform you that your manuscript has been deemed suitable for publication in PLOS ONE. Congratulations! Your manuscript is now being handed over to our production team.

Kind regards, 

on behalf of

Dr. Victor C Huber 

Academic Editor

PLOS ONE